# Evaluation and Comparison of Hybrid Wing VTOL UAV with Four Different Electric Propulsion Systems

**Jianan Zong** **, Bingjie Zhu, Zhongxi Hou \*, Xixiang Yang and Jiaqi Zhai**

College of Aerospace Science and Engineering, National University of Defense Technology,
Changsha 410073, China; zja@nudt.edu.cn (J.Z.); zhubingjie@nudt.edu.cn (B.Z.); nkyangxixiang@163.com (X.Y.);
zhaijiaqi99@163.com (J.Z.)
\* Correspondence: hzx@nudt.edu.cn

**Abstract:** Electric propulsion technology has attracted much attention in the aviation industry at present. It has the advantages of environmental protection, safety, low noise, and high design freedom. An important research branch of electric propulsion aircraft is electric vertical takeoff and landing (VTOL) aircraft, which is expected to play an important role in urban traffic in the future. Limited by battery energy density, all electric unmanned aerial vehicles (UAVs) are unable to meet the longer voyage. Series/parallel hybrid-electric propulsion and turboelectric propulsion are considered to be applied to VTOL UAVs to improve performances. In this paper, the potential of these four configurations of electric propulsion systems for small VTOL UAVs are evaluated and compared. The main purpose is to analyze the maximum takeoff mass and fuel consumption of VTOL UAVs with different propulsion systems that meet the same performance requirements and designed mission profiles. The differences and advantages of the four types propulsion VTOL UAV in the maximum takeoff mass and fuel consumption are obtained, which provides a basis for the design and configuration selection of VTOL UAV propulsion system.

**Keywords:** electric propulsion; hybrid-electric; turboelectric; VTOL UAV; electric aircraft; hybrid aircraft

## 1. Introduction

Over the past decade, the concept of UAVs using electricity has captured the public imagination for some or all of their driving forces and has drawn a lot of attention in popular news [1]. Economy and environmental protection are the main reasons [2]. On the other hand, electric propulsion technology provides incredible new degrees of freedom in UAV system integration to achieve unprecedented collaborative coupling of aerodynamics, propulsion, control, and structure [3]. The unique feature of electric propulsion is that the technology is almost scale-free, allowing small motors to be paralleled for fail safe redundancy or distributed on the fuselage without significant effect on efficiency or specific weight [4–7]. Because of this separation between power and propulsion, efficient, compact motor and transmission systems are now available, enabling many different revolutionary UAV configurations [8–11]. These characteristics of electric propulsion are conducive to the conceptual design and performance improvement of VTOL UAV with large differences in propulsion systems between take-off and cruise conditions [12].

With the development of electric VTOL UAV (eVTOL), urban air mobility (UAM) is highly valued, because eVTOL is considered as the most likely candidate to achieve safe and efficient urban transportation, using electric or hybrid-electric propulsion system and advanced technology concept navigation [13]. NASA [14,15], Uber [16] and other organizations put forward ideas and plans for the development of UAM, pointing out that in the future, advanced air flow can bring transformation in many industries (such as transportation, emergency response, and cargo/packaging logistics), trigger tasks beyond air taxi and parcel express delivery, including the safety of security patrol, emergency and fire, and police patrol. Even in emergency situations, life-saving drugs are provided. In

many VTOL configurations, the hybrid wing UAV can use the rotor to generate vertical lifting force to perform VTOL and hover flight, and at the same time, it can use the fixed-wing to generate lifting force to travel at high speed or long distance [17], which is most suitable for UAM. In this paper, hybrid wing VTOL UAV is as the research object, hereinafter referred to as VTOL UAV.

According to the propulsion system architecture, the classification of electric propulsion system mainly includes: all electric propulsion, series/parallel hybrid-electric propulsion and turboelectric propulsion [18]. Due to the limitation of current battery energy density, the battery weight is too large. Hybrid-electric system is considered as a compromise for quite a long time. Before the battery energy density is broken, it can improve the endurance performance of VTOL UAV. However, compared with all electric propulsion, hybrid-electric propulsion still has the problem of carbon emission, and the design and energy optimization are more complex [19]. Although it is generally considered that turboelectric propulsion system is not suitable for small UAV, it provides an opportunity to change the system architecture and retain the advantages of distributed electric propulsion [20], still expected to be applied to VTOL UAVs. The impact of these four electric propulsion systems with different architectures on small VTOL UAVs has not been fully demonstrated. This paper evaluates the four electric propulsion systems by estimating the maximum take-off weight (MTOM) of VTOL UAVs, because the MTOM is the main factor determining the cost of UAV [21]. In addition, to respond to the environmental protection of the propulsion system, fuel consumption is also used as an evaluation factor.

In this study, the MTOM and fuel consumption of four different electric propulsion VTOL UAVs are estimated by giving several mission profiles. The estimation method is based on the initial sizing method of aircraft [21–24] and there are many researches have improved this method to be suitable for hybrid-electric VTOL UAV sizing [19,25–29]. Finally, sensitivity studies on the cruise distance, takeoff altitude, and payload are carried out, which provides insights for the selection and design of small electric propulsion VTOL UAVs.

## 2. Propulsion and Power System Analysis

In order to find a reasonable design process, the structure of propulsion system, energy-saving mechanism and mass estimation model should be analyzed firstly, and then the final design result should be obtained by the top-level design requirements.

### 2.1. Structures of Electric Propulsion System

The hybrid wing VTOL UAV has two independent systems, which are vertical and horizontal propulsion systems, as shown in Figure 1. The UAV takes off with the lifting force generated by the four lifting rotors vertically. At this stage, the horizontal propulsion motor will be turned off, and the performance of the UAV is similar to that of the quadrotor. After reaching the required altitude, the UAV starts to turn on the horizontal propulsion motor to push the propulsion motor to turn to the horizontal flight [30]. Once the cruise speed is reached, the four lift rotors will be turned off, and the UAV will behave like a fixed-wing UAV.

The four electric propulsion structures applied to VTOL UAV is shown in Figure 2. All electric propulsion system mainly consists of battery, motors and propeller/rotors, and all energy is provided by battery. Turboelectric propulsion system consists of turbine generator, motors and propeller/rotors, which is powered by internal combustion engine (ICE). The parallel hybrid-electric propulsion system consists of ICE, battery, motors and propeller/rotors. In the parallel structure, the horizontal and vertical propulsion systems are independent of each other. The series hybrid-electric propulsion system consists of turbine generator, battery, motors and propeller/rotors. Different from the parallel hybrid system, the whole system is connected by electricity. The ICE and battery can work at the same time or in part, but the generator weight is added.

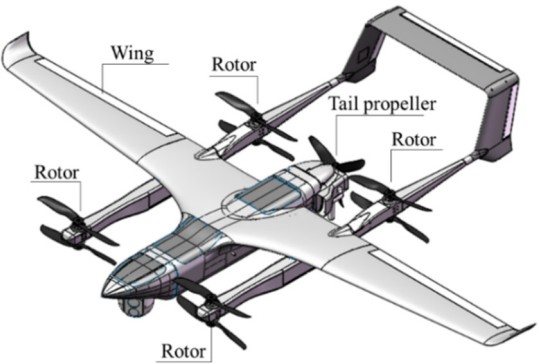

**Figure 1.** Hybrid wing VTOL UAV.

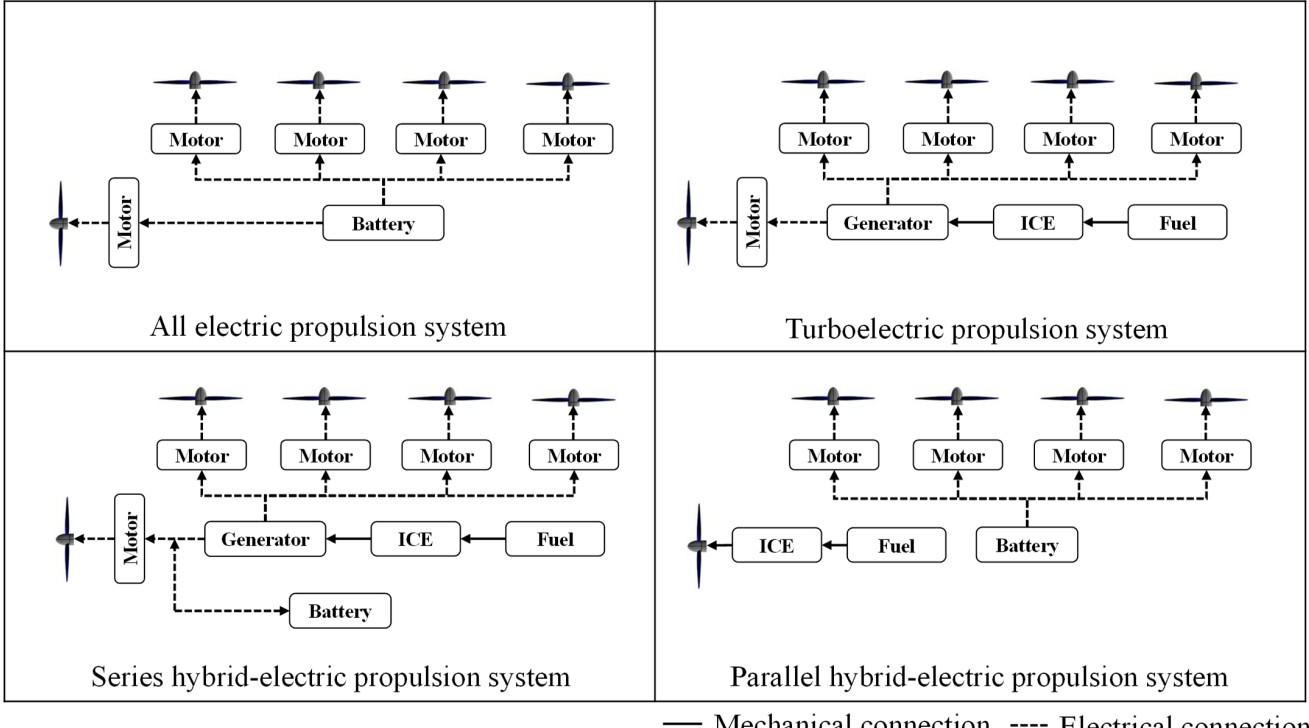

**Figure 2.** Schematic diagram of four electric propulsion configurations.

Through the analysis of four different electric propulsion structures, it can be seen that the working modes of the four systems are different, so it is necessary to analyze the energy-saving mechanism and power distribution of the propulsion system.

### 2.2. Energy Saving Mechanism and Power Distribution

Among the four systems, turboelectric propulsion system and all electric propulsion systems are single energy system. For parallel hybrid system the vertical process and horizontal process are independent. The power demand of vertical process is provided by battery while the power demand of horizontal process is provided by ICE. In series hybrid system, ICE and battery can work at the same time, so the mechanism is more complex than other propulsion system, which needs further analysis.

The series hybrid structure can make ICE always work in the optimal fuel economy area and the inadequate power demand is provided by the battery as the power source. From take-off to landing, ICE can always be stable in the optimal fuel economy area. In the cruise phase, ICE need provide part of the energy to charge the battery. The energy-saving

mechanism of series hybrid-electric system applied to VTOL UAVs can be summarized as three points:

1. Compared with the turboelectric propulsion system, a smaller ICE is needed to meet the power demand of cruise and charging, and the battery provides additional power such as takeoff and climb, so as to improve the load rate of ICE;
2. Make ICE always work in the optimal fuel economy area;
3. In cruise phase, when ICE works in the optimal fuel economy area, the excess energy charges the battery to improve the overall efficiency of the propulsion system.

This energy-saving mechanism and power distribution also provide the most important basis for the power system design, that is, the design parameters of ICE and generator can be determined by the cruise/charging process.

*2.3. Mass Calculation*

2.3.1. Composition of UAV Mass

UAV MTOM can be expressed as the sum of empty mass, payload, and energy mass [22]:

$$MTOM = M_{empty} + M_{payload} + M_{energy}. \tag{1}$$

By expanding empty mass and energy mass, MTOM can be written as:

$$MTOM = M_{struct} + M_{subsyst} + M_{avion} + (M_{prop}^{VTOL} + M_{prop}^{FW}) + M_{payload} + (M_{batt} + M_{fuel}) \tag{2}$$

where payload is the performance requirement given by the top-level design, $M_{prop}^{VTOL}$, $M_{prop}^{FW}$, $M_{batt}$ and $M_{fuel}$ are obtained by calculation. The rest mass is estimated by the weight proportion proposed by Gundlach. The proportion of structural mass is 30–40% of the total weight. Avionics mass proportion account for 5%. The subsystem mass proportion is 5–7%. The MTOM equation can be changed to [22]:

$$MTOM = \frac{M_{prop}^{VTOL} + M_{prop}^{FW} + M_{payload}}{1 - \left( MF_{energy} + MF_{struct} + MF_{subsyst} + MF_{avion} \right)}, \tag{3}$$

where, $MF$ is the mass fraction which represents the ratio of the corresponding subsystem to MTOM.

2.3.2. Power Calculation

The flight conditions of VTOL UAV include takeoff, landing, and cruise. The power demand in fixed-wing mode can be expressed as [31]

$$P_{req} = \frac{\beta W}{\eta_p} \left\{ \frac{V}{\beta} \left( K \frac{\beta^2}{q} \left( \frac{W}{S} \right) + \frac{q C_{D0}}{(W/S)} \right) + \frac{1}{g} \left( \frac{d}{dt} \left( h + \frac{V^2}{2g} \right) \right) \right\}, \tag{4}$$

where, $W$ is the total takeoff weight, $\beta$ is the fuel consumption coefficient which is the ratio of current fuel weight to the initial fuel weight, $q$ is the dynamic head, $C_{D0}$ is the zero-drag coefficient; $K = 1/(\pi e \cdot AR)$ is the drag coefficient due to lift, where $AR$ is the wing aspect ratio, $e$ is the Oswald factor, $h$ is the flight altitude, $dh/dt$ is the rate of climb (ROC), $V$ is the flight speed, and $\eta_p$ is the propeller efficiency. When the cruise is stable, the cruise speed meets the requirements:

$$V = \sqrt{\frac{2 \cdot W}{\rho \cdot S \cdot C_L}}, \tag{5}$$

where $S$ is the wing area and $C_L$ is the lift coefficient.

The typical working conditions of the fixed-wing aircraft in the rotor mode include vertical takeoff and landing, hover, and transition. The calculation formula of propeller/rotor working conditions can be written as [32]:

$$P_{req} = \frac{\beta W}{2FM} \frac{T}{W} \left\{ \left( \left( \frac{dh}{dt} \right)^2 + 4v_i^2 \right)^{0.5} - \frac{dh}{dt} \right\}. \tag{6}$$

The flight condition of the transition is complex, but the duration time is short. The power demand of the fixed-wing VTOL aircraft being a little higher than the maximum power in the rotor mode is appropriate [33].

### 2.3.3. Mass Estimation of Propulsion and Power System

In the process of initial sizing, in order to simplify the model, the mass of each component is usually estimated by power density and energy density, and the complex out of work characteristics are replaced by efficiency. According to the research of Ugur Cakin et al. [27], the technical evaluation of the electric propulsion system is shown in Table 1.

**Table 1.** Technology assessment of hybrid electric systems. Data from [27].

| Parameters | Min | Max |
|---|---|---|
| Electric Motor Efficiency | 0.86 | 0.98 |
| ICE Efficiency | 0.2 | 0.4 |
| Generator Efficiency | 0.86 | 0.98 |
| Battery Efficiency | 0.8 | 0.99 |
| Battery Power Density (kW/kg) | 0.35 | 0.8 |
| Battery Energy Density (Wh/kg) | 151 | 260 |
| ICE Power Density (kW/kg) | 0.25 | 3 |
| Electric Motor Power Density (kW/kg) | 3 | 5 |

The mass of propeller/rotors can be calculated by the formula proposed by Roskam [34]:

$$M_{prop} = 6.514 \cdot 10^{-3} \cdot K_{materail} \cdot K_{prop} \cdot \eta_{props} \cdot n_{blades}^{0.391} \cdot \left( \frac{D_{prop} \cdot P_{\max}}{1000 \cdot n_{prop}} \right)^{0.782}. \tag{7}$$

The recommended value of the coefficient are: $K_{prop}$ = 15, $K_{material}$ = 0.6, where $D_{prop}$ is the diameter of propeller/rotors, which can be expressed as [22]:

$$D_{prop} = K_p \cdot \sqrt[4]{P_{mot}}. \tag{8}$$

The coefficient $K_p$ is 0.1072, 0.0995, and 0.0938 for different blade numbers 2, 3, and 4, respectively.

The energy mass is divided into two parts: battery and fuel, which mainly depends on the mission profiles. It should be noted that the battery needs to meet both power and energy constraints, so the final battery mass is

$$M_{batt} = \max \left\{ \frac{P_{batt,\max}}{\eta_b \cdot PD_{batt}}, \frac{E_{batt,\max}}{\eta_b \cdot ED_{batt}} \right\}, \tag{9}$$

where $\eta_b$ is the battery efficiency, $PD_{batt}$ is the battery power density, $ED_{batt}$ is the battery energy density.

### 3. Initial Sizing Method

Aircraft design is an optimization task. This optimization design problem can be summarized in Table 2. "everything optimization" needs to be simplified in the initial size

determination stage. The size of the fixed-wing UAV is initially determined, and only the highly relevant variables are selected for optimization design [19].

**Table 2.** Design optimization problem Data from [19].

| Minimize | Design Variables | Constraints |
|---|---|---|
| MTOM, cost, fuel mass . . . | Maximum output power of propulsion components, wing area . . . | Performance, aerodynamic characteristics, mechanical characteristics, and stability . . . |

Traditionally, the problem of initial sizing can be described as follows: minimizing MTOM by changing power weight ratio (P/W) and wing loading (W/S), while meeting performance constraints and mission requirements. These parameters are selected because the powertrain and wing area are the main design factors for conventional aircrafts. Therefore, P/W and W/S are selected to minimize the optimization objective [19], and fuel consumption is also an important factor.

*3.1. Initial Sizing Process*

The initial sizing method is an iterative process, as showed in Figure 3. According to the top-level design requirements and a given $MTOM_0$. First, select the optimal design point by analyzing the performance constraints of fixed-wing mode. Thus, the P/W and W/S are obtained. Then, calculating the power demand of the flight process. The mass of propulsion and power system is obtained through the power-mass relationship of propulsion and power system components while the energy mass can be calculated with a given mission profile. The new $MTOM_{new}$ is calculated through the empirical mass proportion as the next input value until the absolute difference between the two adjacent iteration results is allowed, the design process is completed.

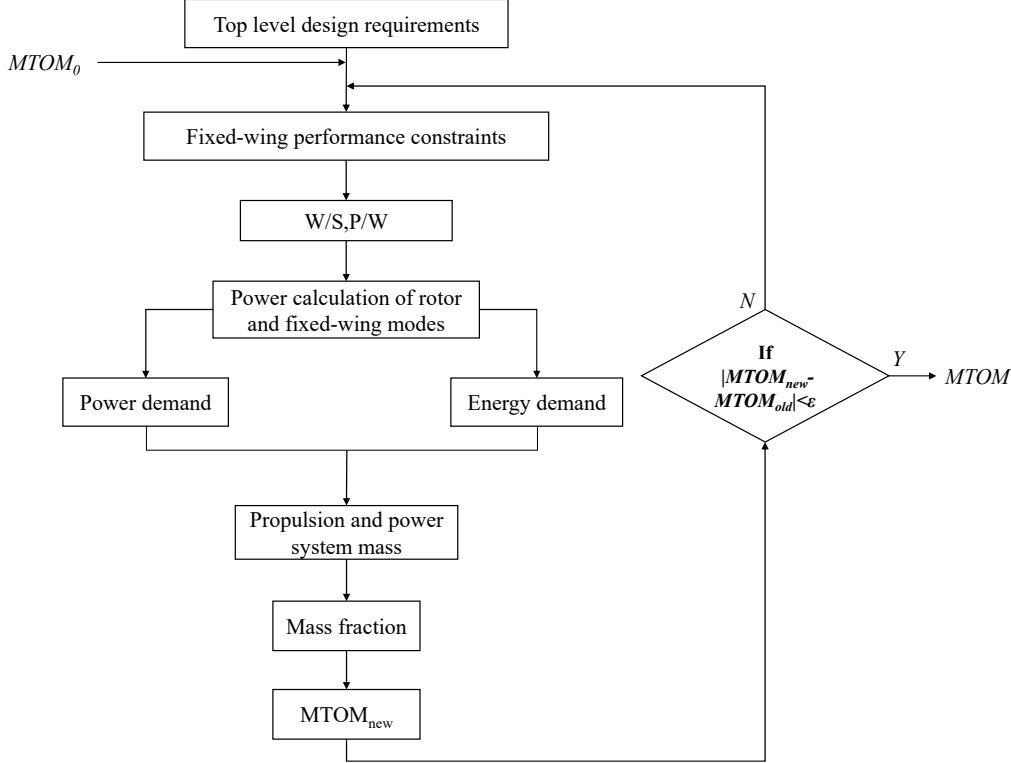

**Figure 3.** Sizing process.

It should be noted that when the iterative process does not converge, the current technical level of propulsion and power system cannot meet the top-level design requirements.

### 3.2. Fixed-Wing Constraint Analysis and Design Point Selection

The traditional design of fixed-wing aircraft includes four constraints: takeoff distance, cruise speed, climb rate, and stall speed [19]. For VTOL UAV, vertical takeoff does not belong to the fixed wing mode, so the constraints of takeoff distance need not be considered in this process, and the constraints of VTOL process will be considered in rotor mode. With all the constraints, the design point and the corresponding propulsion system can be determined according to the design objective. Cruise constraints in fixed-wing mode are given below. These are based on Gudmundsson's equation [21]:

$$c_D = c_{D0} + k \cdot c_L{}^2. \tag{10}$$

Since the output of piston engine and motor is power rather than thrust, it is necessary to convert thrust to power ratio, By assuming the propeller efficiency [21]:

$$P = \frac{T \cdot v}{\eta_P}, \tag{11}$$

where $P$ is power, $T$ is thrust, and $v$ is flight speed, $\eta_p$ is the current propeller efficiency. Cruise constraints can be expressed as:

$$\left(\frac{T}{W}\right)_{cruise} \geq \frac{q}{(W/S)} \cdot C_{D0} + k \cdot \frac{1}{q} \cdot W/S, \tag{12}$$

$$q = \frac{1}{2} \cdot \rho \cdot v. \tag{13}$$

According to the analysis in Section 2.2, for the series hybrid-electric system UAV, ICE needs to charge the battery in the cruise phase. Therefore, there is an additional charging mode constraint. This paper assumes that ICE reserves 20% of the power output to charge the battery in the cruise phase [35]. This constraint only needs to be multiplied by 1.2 on the cruise speed constraint. The climbing constraint of can be expressed as

$$\left(\frac{T}{W}\right)_{climb}^{FW} \geq \frac{ROC}{v_y} + \frac{q}{W/S} \cdot C_{D0} + \frac{k}{q} \cdot (W/S). \tag{14}$$

The optimal climbing speed can be expressed as:

$$v_y = \sqrt{\frac{2}{\rho} \cdot W/S \cdot \sqrt{\frac{k}{3C_{D0}}}}. \tag{15}$$

The following expression is used to determine the T/W required to achieve a given service ceiling, assuming it is where the best rate-of-climb of the airplane has dropped to 0.5 m/s. The upper bound for the aircraft's wing loading is determined by a stall speed requirement in an FW mode. The wing loading required to maintain the stall speed is calculated as:

$$W/S_{stall}^{FW} = \frac{1}{2} \cdot v_{stall}^2 \cdot \rho \cdot C_{L,\max}. \tag{16}$$

In the constraint diagram, the stall speed constraint is the right boundary of the design region. The dimensionless constraint diagram is shown in Figure 4.

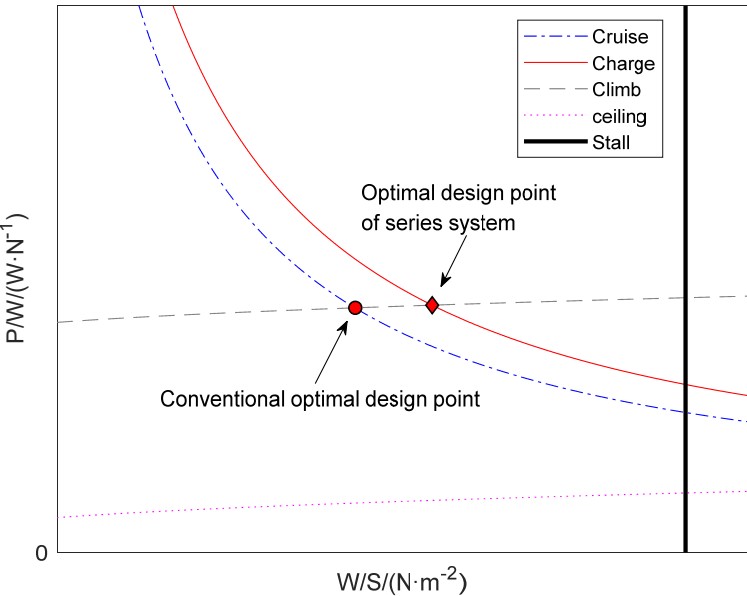

**Figure 4.** Design space of point performance constraints in fixed-wing mode.

The diagram also features the optimal design point, which requires the least power or thrust to meet all applicable requirements. This point results in minimum power required and this usually means a power plant that is less expensive to acquire and operate [21]. The red dot is the design point of all electric propulsion system, turbine electric propulsion system, and parallel propulsion system. Since the series hybrid system needs to charge the battery in the cruise phase, the design point is the intersection of the charging constraint and the climbing constraint, which is the red diamond point.

## 4. Numerical Results

In this section, the same performance requirements and three different mission profiles are given. We estimate the size of four kinds of electric propulsion systems respectively, and compare their weight differences in different cases. Performance requirements and mission profiles parameters are shown in Tables 3 and 4, respectively.

**Table 3.** Performance requirements.

| Performance | Value |
| --- | --- |
| Cruise speed | 30 m/s |
| *ROC* | 3 m/s |
| Service ceiling | 1000 m |
| Stall speed | 12.5 m/s |
| Payload | 10 kg |

**Table 4.** Mission profile paraments.

| Cases | Takeoff Altitude (m) | Cruise Distance (km) |
| --- | --- | --- |
| Case 1 | 500 | 30 |
| Case 2 | 1500 | 100 |
| Case 3 | 3000 | 500 |

### 4.1. Case Studies

The MTOM and the mass of each part of the UAVs with case 1 are shown in Figure 5. The MTOM from large to small are all electric UAV, turboelectric UAV, parallel hybrid-electric UAV, and series hybrid-electric UAV, and the values are 41.35 kg, 24.26 kg, 23.98 kg,

and 20.89 kg respectively. The fuel consumption from large to small are turboelectric UAV, series hybrid-electric UAV, parallel hybrid-electric UAV, and all electric UAV. The values are 0.156 kg, 0.135 kg, 0.11 kg and 0 kg respectively. The MTOM of all electric UAV is the largest and much larger than other three propulsion system UAV, but the advantage of all electric propulsion is zero emission. Parallel hybrid-electric UAV has the least fuel consumption, because the ICE will work only in cruise phase. The fuel consumption of turboelectric UAV is the largest, and MTOM of turboelectric UAV is also larger than that of series and parallel hybrid-electric system. In case 1, the series hybrid-electric UAV holds the smallest MTOM.

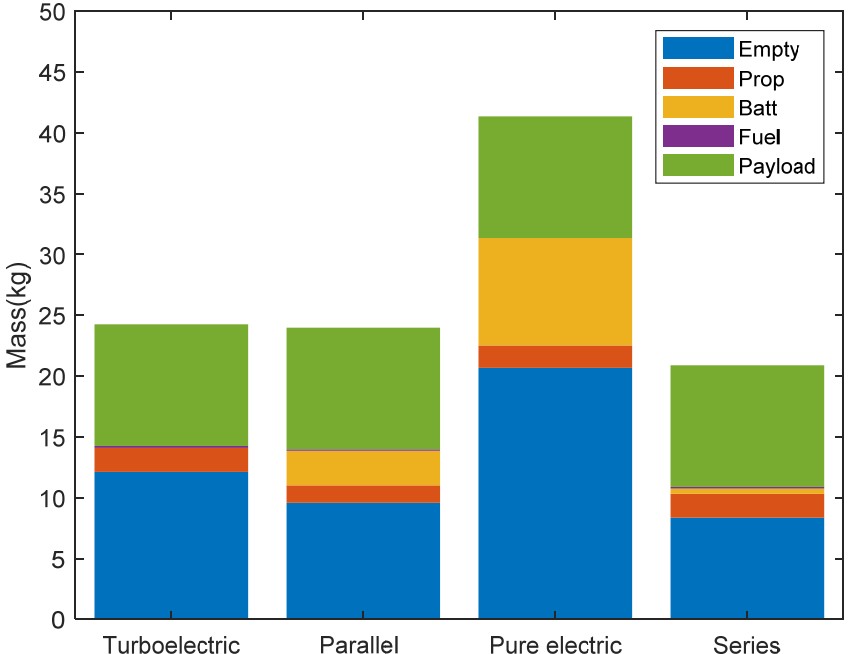

**Figure 5.** Case 1: MTOM and mass of other parts of UAVs.

For VTOL UAV with electric propulsion system in case 2, the propulsion system cannot meet the requirements, because the iterative process will not converge. Therefore, Figure 6 only shows the MTOM of the other three propulsion system UAVs and the mass of each part. At this time, the MTOM from large to small are parallel hybrid-electric UAV, turboelectric UAV, and series hybrid-electric UAV, the values are 28.32 kg, 25.16 kg and 21.57 kg respectively. The fuel consumption of parallel hybrid-electric UAV and series hybrid-electric UAV are the same, which are 0.45 kg, while turboelectric UAV is 0.52 kg. The fuel consumption of series hybrid-electric system and parallel hybrid-electric system is minimum. Therefore, the series hybrid-electric propulsion system is the preferred propulsion system in case 2.

In case 3, only turboelectric and series hybrid-electric UAV can meet the requirements. As shown in Figure 7, compared with turboelectric UAV, series hybrid-electric UAV has the best performance in MTOM and fuel consumption. The MTOM was 26.34 kg and 30.54 kg, respectively, and the fuel consumption was 2.34 kg and 2.73 kg, respectively.

After the calculation of three cases, we integrate the results into Table 5. The "/" indicates that the iterative process does not converge.

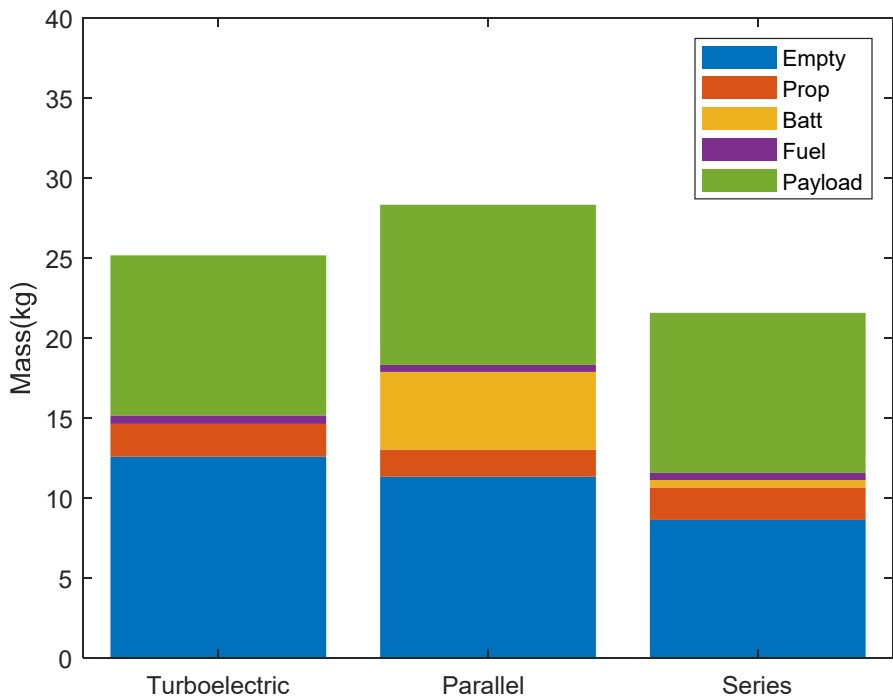

**Figure 6.** Case 2: MTOM and mass of other parts of UAVs.

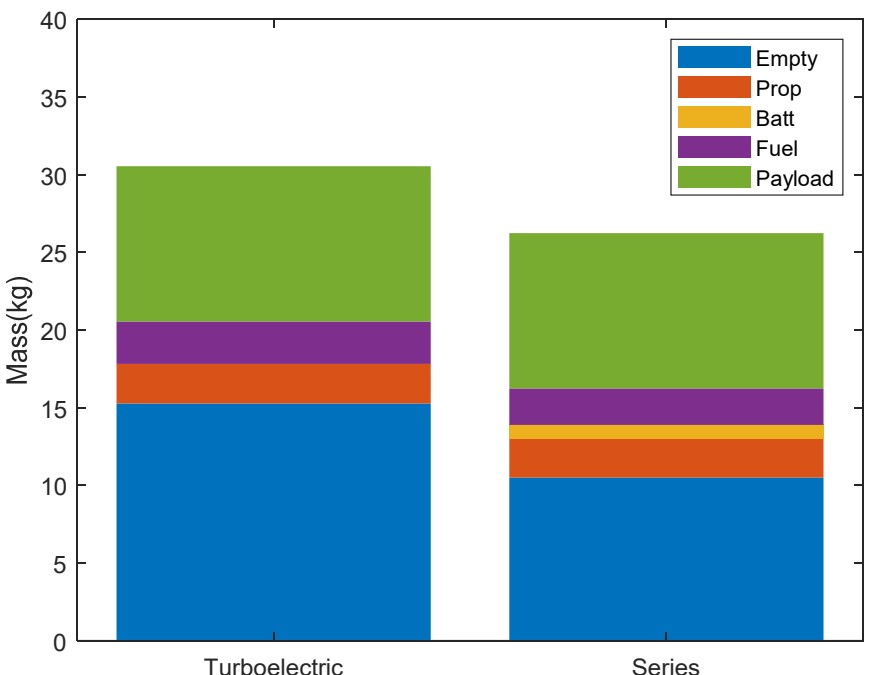

**Figure 7.** Case 3: MTOM and mass of other parts of UAVs.

**Table 5.** Mission profile paraments.

| Types | Case 1 | | Case 2 | | Case 3 | |
|---|---|---|---|---|---|---|
| | **MTOM (kg)** | **Fuel (kg)** | **MTOM (kg)** | **Fuel (kg)** | **MTOM (kg)** | **Fuel (kg)** |
| All electric | 41.35 | 0 | / | / | / | / |
| Series hybrid | 20.89 | 0.135 | 21.57 | 0.45 | 26.34 | 2.34 |
| Parallel hybrid | 23.98 | 0.11 | 28.32 | 0.45 | / | / |
| Turboelectric | 24.26 | 0.156 | 25.16 | 0.52 | 30.54 | 2.73 |

Through the analysis of three cases, it can be concluded that the maximum flight profile of all electric VTOL UAV is the poorest, and the MTOM is the largest, but with the advantage of zero fuel consumption; The series hybrid-electric VTOL UAV always holds the smallest MTOM. Except case 1, the fuel consumption of series hybrid-electric VTOL UAV is also the minimum, so the advantage is very prominent. The maximum flight profile of parallel hybrid-electric VTOL UAV is inferior to that of turboelectric and series hybrid-electric VTOL UAV, but in a certain flight profile, it still has relatively low fuel consumption and MTOM. However, with the expansion of flight profile, turboelectric propulsion has more advantages than parallel system.

*4.2. Sensitivity Study*

In order to further study the influence of key performance parameters on the design results of VTOL UAV, this section studies sensitivity of the three parameters, which are takeoff altitude, cruise distance, and payload, respectively. The sensitivity study is based on case 1, which focuses on the result changes of UAV MTOM and fuel consumption, because they are highly related to cost and environmental protection, respectively. The sensitivity analysis process of each parameter is univariate, that is, other variables are frozen to the one from case 1.

4.2.1. Takeoff Altitude Sensitivity

The effect of takeoff altitude on MTOM and fuel consumption are shown in Figure 8. For MTOM, takeoff altitude has the greatest effect on all electric UAV and parallel hybrid-electric UAV, but has relatively little effect on turboelectric UAV and series hybrid-electric UAV. Moreover, the maximum takeoff altitude of all electric UAV is minimum, only 1100 m, followed by parallel hybrid-electric UAV, which maximum takeoff altitude is 2500 m. Both turboelectric and series hybrid-electric UAV exceed 5000 m, of which series hybrid-electric UAV holds the minimum MTOM.

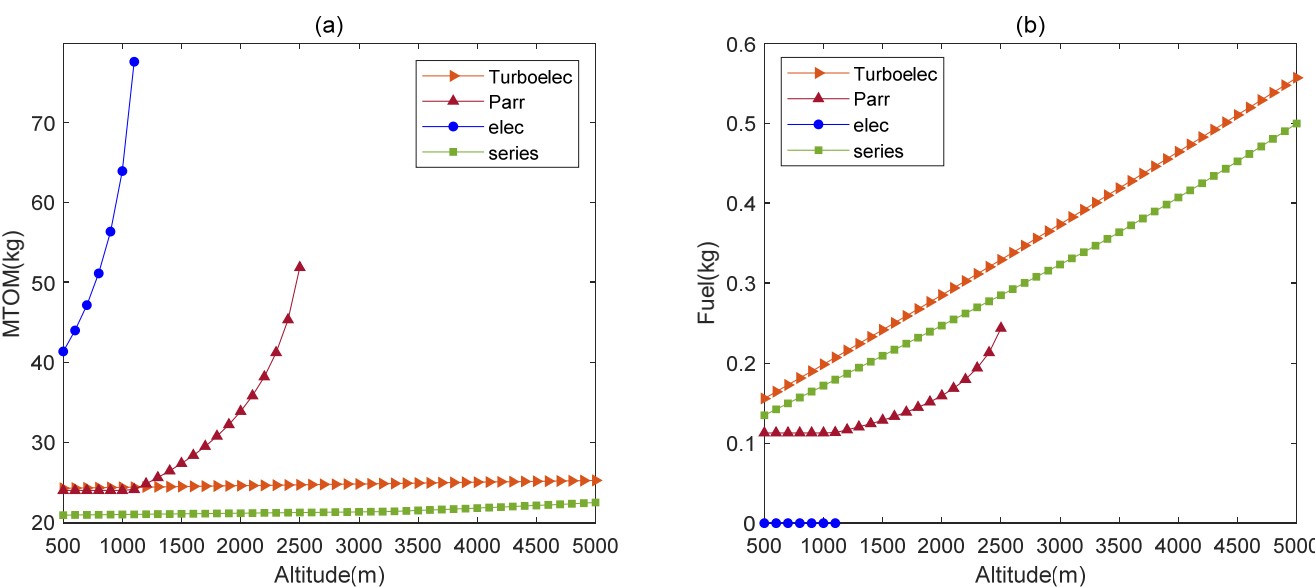

**Figure 8.** Effect of takeoff altitude: (**a**) MTOM; (**b**) fuel consumption.

In terms of fuel consumption, all electric propulsion has zero emission, and parallel hybrid-electric propulsion takes the second place. However, the takeoff altitude of these two of propulsion UAV is limited. The fuel consumption of the series hybrid-electric UAV is less than that of the turboelectric UAV, and the turboelectric UAV has the largest fuel consumption.

### 4.2.2. Cruise Distance Sensitivity

The effect of cruise distance on MTOM and fuel consumption is shown in Figure 9. The change of MTOM of all electric propulsion UAV is much higher and the maximum cruise distance is much smaller than that of UAV of other three propulsion systems. The relationship between MTOM and fuel consumption of parallel hybrid-electric UAV and turboelectric UAV are almost the same as that of cruise distance. Except the advantage of zero emission of all electric UAV in the range of 50 km cruise distance, the series hybrid-electric UAV has the lowest fuel consumption.

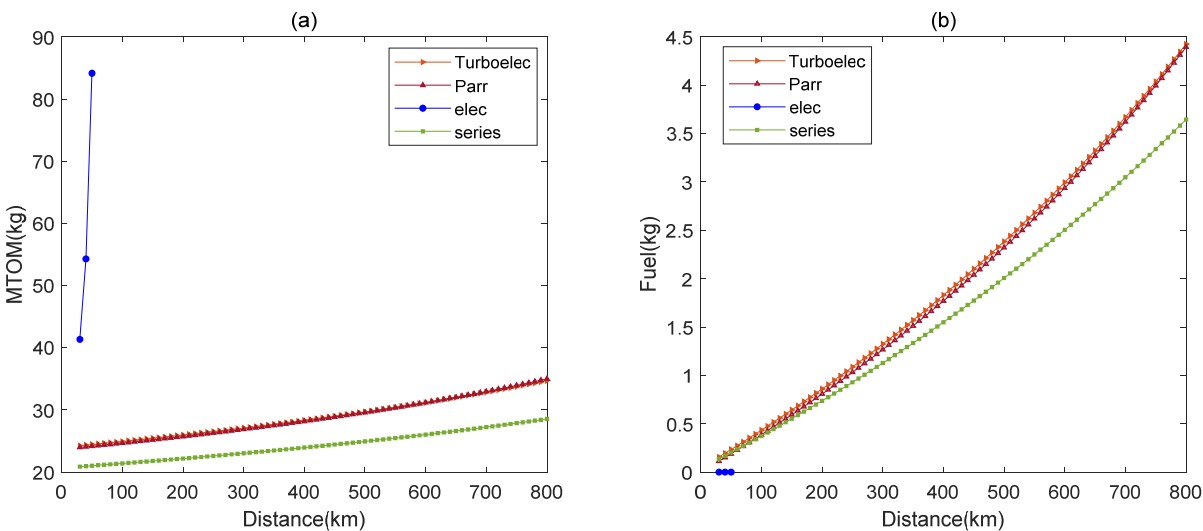

**Figure 9.** Effect of cruise distance: (**a**) MTOM; (**b**) fuel consumption.

### 4.2.3. Payload Sensitivity

The effect of payload mass on MTOM and fuel consumption is shown in Figure 10. Payload has the greatest effect on MTOM of all electric UAV, which is much higher than the other three propulsion UAVs. When the payload is light, the difference of MTOM between series hybrid-electric UAV, parallel hybrid-electric UAV, and turboelectric UAV is small, and the fuel consumption of parallel hybrid-electric UAV is the smallest. However, with the increase of payload, parallel hybrid-electric UAV no longer has advantages in MTOM. Series hybrid-electric UAV is superior to turboelectric UAV in MTOM and fuel consumption.

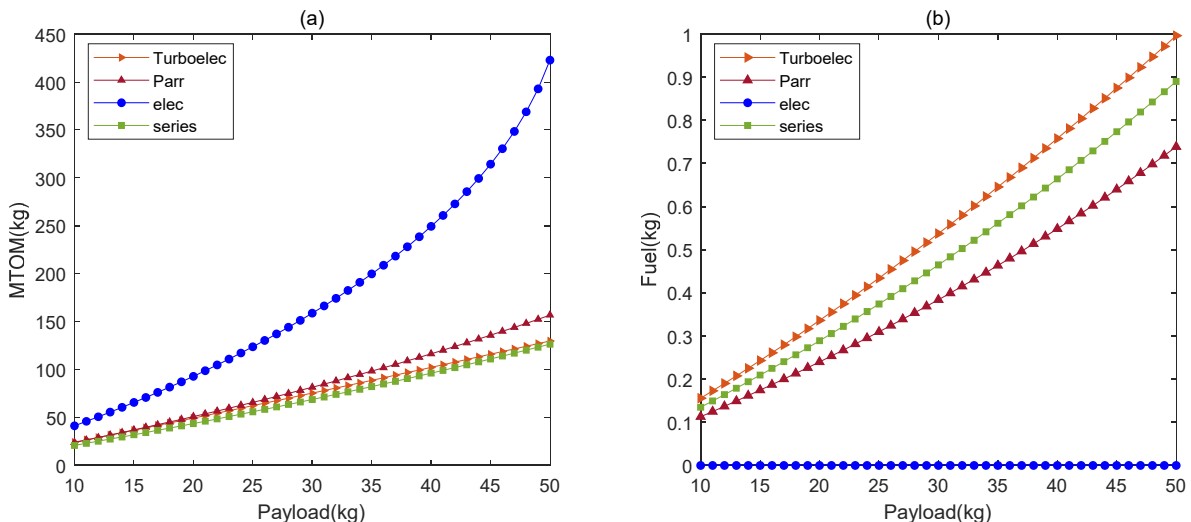

**Figure 10.** Effect of payload: (**a**) MTOM; (**b**) fuel consumption.

It is worth noting that the effect of payload on MTOM is much higher than takeoff altitude and cruise distance, and increasing the mass of payload can make the UAV iterate to a much larger MTOM but still converge.

## 5. Conclusions

By analyzing VTOL UAV MTOM and fuel consumption, this paper evaluates and compares the potential of four different configurations of electric propulsion systems in VTOL UAV. We find that with the current level of battery technology, the all electric propulsion VTOL UAV has the disadvantages of overweight and small flight area. Hybrid-electric propulsion or turboelectric propulsion can greatly improve the performance and manufacturing costs of UAV. The series hybrid propulsion system is the most promising electric propulsion configuration for hybrid wing UAV. With the increase of battery energy or power density, the disadvantages of all electric propulsion system will be gradually improved. In the future, for a certain flight area, what level or how long can the all-electric propulsion system be considered to have better comprehensive performance, needs further research.

**Author Contributions:** Conceptualization, J.Z. (Jian Zong) and Z.H.; methodology, J.Z. (Jian Zong); software, J.Z. (Jian Zong); validation, J.Z. (Jian Zong), B.Z. and J.Z. (Jiaqi Zhai); formal analysis, J.Z. (Jian Zong); investigation, J.Z. (Jian Zong); resources, Z.H.; data curation, B.Z.; writing—original draft preparation, J.Z. (Jian Zong); writing—review and editing, J.Z. (Jian Zong); visualization, J.Z. (Jian Zong); supervision, X.Y.; project administration, X.Y.; funding acquisition, X.Y. All authors have read and agreed to the published version of the manuscript.

**Funding:** This research was funded by the National Nature Science Foundation of China (52172410) and National Natural Science Foundation of China (61703414).

**Institutional Review Board Statement:** Not applicable.

**Informed Consent Statement:** Not applicable.

**Data Availability Statement:** All data generated or analyzed during this study are included in this article.

**Conflicts of Interest:** The author declares no conflict of interest.

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
