# Peer review of "Evaluation and Comparison of Hybrid Wing VTOL UAV with Four Different Electric Propulsion Systems"

_aerospace, doi:10.3390/aerospace8090256_

Round 1

Reviewer 1 Report

The paper presents a comparative study of four different configuration of electric propulsion system for fixed wing VTOL UAV. In general the contents of a paper is interesting and easy to understand. The decriptions are clear as well as the language; In my opinion the approach for the analysis is correct and the major issue is the significance of a content. From the general point of view, the conclusions are based on the limited set of data. In the scenarion analysis, both parameters (take off altitude and cruise distance) changes. Wouldn't it be good to compare the influence the one parameter at once and then change the other adding the next level of analysis? It is also not clear how the performance parameters were chosen for the analysis and how they influences the results at the end. Sensivity analysis in based on case 1 - but what is that mean? During the sensivity analysis both parameters (take off altitude and cruise distance) changes so how thas in based on case 1? Is it mean that during the analysis of take of altitude the cruise distance value is frozen to the one from case 1? That should be explaind more clear. On the other hand the more general results would be expected in the form of e.g. 3D maps when both parameters are changing and presented on the same graph. 

Additionaly, there are some spotted issues:

  1. all the symbols and abbreviation should be clarified in the text or dedicated section;
  2. In the abstract: "Limited by battery energy density, series/ parallel hybrid-electric propulsion and turboelectric propulsion are considered to be applied to VTOL UAVs to improve performances. In this paper, the potential of these four configurations of electric propulsion systems for small VTOL UAVs is evaluated and compared.” 3 configurations are listed and a second after that four are mentioned; There is lack of all electric configuration in my opision;
  3. Authors using a parameter of maximum take-off weight and they choose the abbreviation of MTOM. In the opinion of a reviewer that’s inconsistency and should be “maximum take-off mass” of abbreviation MTOW;
  4. In a section 2.2 authors claims that: “The energy-saving mechanism of ICE applied to the VTOL UAV can be summarized as two points” and then listing three points – inconsistency again;
  5. In a section 2.2 point 1 in a list: “A smaller ICE…” – smaller comparing to what?
  6. In section 2.3.3 coefficient Kmaterail – is that the typo?
  7. In the same section below: “recommended values on the coefficients”. Recommended based on what? How those values changes for other cases? What is the range of those coefficients?
  8. What is Kp in Dprop expression? Where are the values comes from?
  9. Caption of Figure 2 should be extended;
  10. In equation 10 authors take into account the propeller efficiency. Is there any gear box between the engine and the propeller assumed? Is the propeller direct driven by the piston engine?
  11. In section 3.2 authors “assumes that ICE reserves 20% of the power output to charge the battery in the cruise phase.” What is the source of that assumption? Why not 10 or 30%? How would that influence the analysis or results?
  12. Table 3 – What is the source for those requirements? How they correspond to the currently used or planned UAVs?
  13. In the first paragraph of section 4.1 the information: “The MTOM of all electric UAV is the largest and much larger than other three propulsion system UAV, but the advantage of all electric propulsion is zero emission” is doubled;
  14. The values of fuel masa in case 1 are surprisingly low – could you comment on that?
  15. In section 4.1 statement: “The fuel consumption of parallel hybrid-electric UAV  and series hybrid-electric UAV are the same, which are 0.45kg and 0.53kg respectively” is not clear for me;
  16. In section 4.1 statement: “In case 2, the all electric propulsion system cannot meet the mission performance requirements” is doubled;
  17. That would be very helpful to present the results from cases 1,2,3 also in the form of table;
  18. In section 4.1 statement: “Through the analysis of three cases, it can be concluded that the maximum flight profile of all electric VTOL UAV is the poorest, and the MTOM is the largest, but with the advantage of zero fuel consumption” – I wonder is that a really an advantage? What is the reason for that? Is that the environmental one? Did you considered the process of battery production, charging and utlilization in your analysis?
  19. Conclusion section – In my opinion both – analysis and conclusions, should be more general (not focused on the analysis of particular mission profiles) to be applicable for wider audience;

Reviewer 2 Report

The paper is describing a comparison evaluation in the fuel consumption of a UAV-type vertical take-off and landing craft. The theme of the work seems quite interesting, but of course, some minor improvements would lead to higher clarity and quality of the paper.

1- I would suggest that the authors reconsider the title of the paper (which does not seem appropriate currently). The title by now seem a full sentence and too long. Better to make it to the point and short.

2- Once you review the manuscript and you find some minor typos and corrections which need to be taken care of.

3- The cited references also can be improved and comprised of a wider range of publications from top-level journals on the VTOL itself.

Round 2

Reviewer 1 Report

Dear Authors, 

I found your responses and corrections sufficiently good;

During the second reading I found two small issues:

  • you cite the same book (but different edition) [22] and [36] - is it necessary?
  • some references in the text are without a "space" between the number and followed word and at leat one is put as a superscript e.g. [22];

I will recommend the paper for publication after those corrections;

Author Response

We appreciate and marvel at your careful correction. We have added spaces before the references one by one and corrected the superscripts according to your requirements. After our verification, the reference [36] has corrected to the same book of the reference [22].
